# Identification of Antioxidant Peptides Derived from Tilapia (*Oreochromis niloticus*) Skin and Their Mechanism of Action by Molecular Docking

**DOI:** 10.3390/foods11172576

**Published:** 2022-08-25

**Authors:** Yueyun Ma, Dandan Zhang, Mengqi Liu, Yingrou Li, Rui Lv, Xiang Li, Qiukuan Wang, Dandan Ren, Long Wu, Hui Zhou

**Affiliations:** 1College of Food Science and Engineering, Dalian Ocean University, Dalian 116023, China; 2National R&D Branch Center for Seaweed Processing, Dalian 116023, China; 3Key Laboratory of Aquatic Product Processing and Utilization of Liaoning Province, Dalian 116023, China; 4Collaborative Innovation Center Jointly Established by Ministries and Ministries for Key Technologies of Deep Processing of Marine Food, Dalian Polytechnic University, Dalian 116034, China

**Keywords:** molecular docking, antioxidant, peptide, free radical, tilapia

## Abstract

Antioxidants, which can activate the body’s antioxidant defence system and reduce oxidative stress damage, are important for maintaining free radical homeostasis between oxidative damage and antioxidant defence. Six antioxidant peptides (P1–P6) were isolated and identified from the enzymatic hydrolysate of tilapia skin by ultrafiltration, reversed-phase high-performance liquid chromatography (RP-HPLC) and liquid chromatography–tandem mass spectrometry (LC–MS/MS). Moreover, the scavenging mechanism of the identified peptides against DPPH (2,2-Diphenyl-1-picrylhydrazyl) and ABTS (2-azido-bis (3-ethylbenzothiazoline-6-sulfonic acid) was studied by molecular docking. It was found that Pro, Ala and Tyr were the characteristic amino acids for scavenging free radicals, and hydrogen bonding and hydrophobic interactions were the main interactions between the free radicals and antioxidant peptides. Among them, the peptide KAPDPGPGPM exhibited the highest DPPH free radical scavenging activity (IC_50_ = 2.56 ± 0.15 mg/mL), in which the hydrogen bond between the free radical DDPH and Thr-6 was identified as the main interaction, and the hydrophobic interactions between the free radical DDPH and Ala, Gly and Pro were also identified. The peptide GGYDEY presented the highest scavenging activity against ABTS (IC_50_ = 9.14 ± 0.08 mg/mL). The key structures for the interaction of this peptide with the free radical ABTS were identified as Gly-1 and Glu-5 (hydrogen bond sites), and the amino acids Tyr and Asp provided hydrophobic interactions. Furthermore, it was determined that the screened peptides are suitable for applications as antioxidants in the food industry, exhibit good water solubility and stability, are likely nonallergenic and are nontoxic. In summary, the results of this study provide a theoretical structural basis for examining the mechanism of action of antioxidant peptides and the application of enzymatic hydrolysates from tilapia skin.

## 1. Introduction

The human body continuously generates free radicals when stimulated by external environments, such as respiration, external pollution, radiation exposure and other factors. Cells and tissues are broken down when free radicals accumulate in the human body, which is followed by a negative effect on metabolic function [1]. Excessive free radicals are almost always associated with cancer [2], ageing [3] or other diseases. Antioxidants have been shown to be effective in preventing and treating these diseases. They reduce oxidative stress damage by maintaining the balance between oxidative damage and antioxidant defence [4]. While exogenous active peptides are an important source of antioxidants, the enzymatic treatment of macromolecular proteins is the most commonly used method to obtain antioxidant peptides [5,6]. For example, the swim bladder protein of miiuy croaker (*Miichthys miiuy*) was hydrolysed by an alkaline protease to obtain the antioxidant peptides FPYLRH and GIEWA [7]. Antarctic krill were treated with pepsin, LKPGN and LQP from enzymatic hydrolysis products and showed antioxidant capacity [8]. It has been reported that various mechanisms such as radical scavenging, metal chelation and reduction of oxidizing species were summarized, among which the scavenging of radicals is considered to be one of the important ways to prevent oxidation [4,9]. At present, it is generally believed that the amino acid sequence and the spatial structure of the peptide chain have a greater impact on antioxidant capacity [10,11]. However, the molecular mechanism of action for amino acids is still unclear. How amino acid structures are critical remains questionable. Therefore, further studies on the mechanism by which antioxidant peptides scavenge free radicals are needed.

It is well known that molecular structure is the material basis for the expression of peptide activity. Studies on the structure–activity relationship of antioxidant peptides are essential to clarify the mechanism by which active peptides scavenge free radicals. Molecular docking [12] is a simulation method based on key theoretical foundations that can predict the binding mode and interaction force between peptides and free radicals, thereby explaining the mechanism of action for free radicals at the molecular level. The configuration of the polypeptide molecule with a known amino acid sequence is adjusted and optimized by MM2 to minimize the energy, and MM2 is a molecular mechanics method developed by Allinger, N.L. used for energy optimization of the molecular configurations [13]. The docking process is a combination of grid-based energy evaluation and a search for torsional freedom. Scans are first performed using different types of atoms as probes to calculate the grid energy. A conformational search is then performed within a specified range, and the results are finally scored and ranked [14]. The global energy minimum is obtained in the binding energy of the ligand and the receptor, thus the optimal docking result can be obtained between the peptide and the specific free radical molecule. In recent years, this technology has been applied to study the mechanism of action of active peptides [15,16]. Wen, C. et al. [16] discovered the antioxidant mechanism of watermelon seed peptide RDPEER to the free radicals DPPH (2,2-Diphenyl-1-picrylhydrazyl) and ABTS (2-azido-bis (3-ethylbenzothiazoline-6-sulfonic acid) diammonium salt) by molecular docking. ATVY [15] from black shark skin exhibited satisfactory scavenging activity against ABTS. Tyr at the N-terminus is a key site for interactions with free radicals as determined by molecular docking analysis, which confirms the effectiveness of this method. Therefore, molecular docking is currently an effective tool to reveal the underlying mechanism of antioxidant peptides to scavenge free radicals, and studying the mechanism of action of active peptides by molecular docking would promote the development of active peptides.

Nile tilapia (*Oreochromis niloticus*), a cichlid family, tilapia fish, is among the most popular farmed freshwater fish in the world [17,18]. As the by-product of the production of tilapia fillets, tilapia skin is usually discarded as waste, causing environmental pollution and wasting resources [19]. Tilapia skin contains abundant protein content (89.15%, dry basis), and the peptides from fish skin hydrolysates have been demonstrated to be biologically active [20,21]. The antioxidant capacity of tilapia skin peptide (TSP) was confirmed by chemical-based in vitro tests [22,23,24], but the specific antioxidant mechanism between TSP and oxidative radicals still remains undefined.

Therefore, this work focused on antioxidant peptides derived from tilapia skin and their mechanism of action by molecular docking, as shown in Figure 1. The purpose of our study was to (1) screen out new antioxidant peptides derived from tilapia skin protein hydrolysates; (2) identify the characteristic amino acids of antioxidant peptides in free radical scavenging (DPPH, ABTS) reactions; and (3) discover the dominant conformation and interaction mechanism of antioxidant peptides and free radicals by molecular docking. The outcome of this study is expected to provide a structural theoretical basis for the mechanistic research direction of tilapia skin antioxidant peptides. At the same time, the developed active peptides can be used in health food, precision medicine, traditional Chinese medicine, molecular biology and other fields.

## 2. Materials and Methods

### 2.1. Materials

Nile tilapia skins were obtained from Yantai (Shandong, China). Alkaline protease (EC 3.4.21.14) was purchased from Yuanye Biotechnology Co., Ltd. (Shanghai, China). Glutathione (GSH), DPPH and ABTS were purchased from Maclean Biochemical Technology Co., Ltd. (Shanghai, China). HPLC-grade trifluoroacetic acid (TFA), formic acid (FA), acetonitrile (ACN) and methanol were purchased from Merck (Darmstadt, Germany). All other reagents used were of analytical grade.

### 2.2. Preparation of Tilapia Skin Peptide (TSP)

The tilapia skin peptide (TSP) was prepared by enzymolysis. The enzymatic hydrolysis of tilapia skin was carried out according to the pre-experimental optimization method in order to obtain the enzymatic hydrolysate with the maximum antioxidant capacity in vitro. The skin was cut into pieces (approximately 1 × 1 cm^2^), washed with running tap water and dried by lyophilization. According to a 1:50 (g/mL) material–liquid ratio, distilled water was added to the lyophilized tilapia skin stock and was adjusted to pH 10 by 0.1 mol/L sodium hydroxide solution. The mixture was treated with alkaline protease at 2.51% (enzyme to tilapia skin ratio) of the enzyme added at 51 °C. The mixture was maintained under constant stirring for 9 h and then boiled for 15 min to inactivate the protease. The supernatant was collected after centrifugation at 10,000 r.p.m. for 15 min and then lyophilized, and the peptide powder was stored at −20 °C until use.

### 2.3. Separation and Purification of TSP

#### 2.3.1. Peptide Separation by Ultrafiltration

Ultrafiltration separation of the samples was accomplished at room temperature by ultrafiltration equipment (Millipore, Darmstadt, Germany). The chamber volume of the device was 2 L, and the material of the ultrafiltration membrane was polyethersulfone (PES). The TSP powder was dispersed in distilled water, and the mixture was ultrafiltrated by the ultrafiltration membranes with 5 kDa, 3.5 kDa and 1 kDa molecular weight cut-off (MWCO) values. Thus, four fractions, M1, M2, M3 and M4, with molecular weights >5 kDa, 3.5–5 kDa, 1–3.5 kDa and <1 kDa, respectively, were collected and lyophilized, and the free radical scavenging ability of each fraction to DPPH and ABTS was tested.

#### 2.3.2. Peptide Separation by Reversed-Phase High-Performance Liquid Chromatography

The fraction with the highest antioxidant capacity among M1, M2, M3 and M4 was further purified using a high-performance liquid chromatography (HPLC) system with the reversed-phase preparative column PREP-ODS (H) KIT (20 × 250 mm, Shimadzu). The lyophilized fraction was dissolved in ultra-pure water to obtain 30 mg/mL solutions, and the solutions were loaded onto the column (700 μL loading volume). The flow rate was set to 7 mL/min using 5% ACN (0.1% TFA) and 80% ACN (0.1% TFA) as eluents A and B, respectively. Elution was performed using the following sequence: 0–10 min, 10% eluent B; 10–15 min, 10–80% eluent B; 30–40 min, 80–10% eluent B. The peaks were monitored at 220 nm, and the eluted fractions were collected separately and then freeze-dried. The free radical scavenging ability of the samples was determined.

#### 2.3.3. Identification of Peptides by LC–MS/MS

The liquid chromatography–tandem mass spectrometry (LC–MS/MS) analyses were performed on an LTQ-Orbitrap Elite mass spectrometer (Thermo Scientific, Boston, MA, USA) equipped with a Dionex Ulti Mate 3000 RSLC nano system (Thermo Scientific, Boston, MA, USA) for separation. The peptides were separated on a reverse-phase (RP) C18 analytical column (15 cm × 150 µm i.d.). The lyophilized samples were redissolved in 0.1% FA, and 1 µg of sample was taken for analysis. The flow rate was set at 600 NL/min. Mobile phase: buffer A (100% H_2_O/0.1% FA), buffer B (80% ACN/0.1% FA). The binary separation gradient was employed as follows: 2–8% buffer B for 2 min, 8% to 45% buffer B for 60 min and 45% to 95% buffer B for 3 min. The mass spectrometer was operated in positive ion data-dependent acquisition (DDA) mode. Data were processed with an in-house Mascot server (version 2.4.1) (London, UK) and searched with the UniProt tilapia database.

#### 2.3.4. Peptide Synthesis

Based on the above analysis, selected peptides were synthesized by the solid phase peptide synthesis (SPSS) procedure and purified by RP-HPLC using a C18 column (4.5 × 250 mm, 5 μm). MS data for the synthetic peptide was confirmed using liquid chromatography–mass spectrometry (LC–MS) (Shimadzu LC-MS2020, Kyoto, Japan). The purities of the synthesized peptides were over 95%.

### 2.4. Antioxidant Capacity Assay

The antioxidant capacity of peptides was evaluated by two of the most widely used methods, which are the ABTS and DPPH assays [4].

#### 2.4.1. DPPH Radical Scavenging Activity

The DPPH radical scavenging activity of TSP was determined according to the previous method of Wali, A. et al., with minor modifications [25]. Glutathione (GSH) and the sample were prepared with the test solution with concentration gradients of 2, 4, 6, 8 and 10 mg/mL, respectively. Then, 100 µL of the sample solution was added to 100 µL of DPPH solution (0.05 mg/mL in ethanol) and incubated for 30 min at room temperature in the dark. The absorbance at 517 nm was measured by an enzyme-labelled instrument (Synergy H1M, Beijing, VT, USA). Ethanol instead of the DPPH solution was used as a control, and ethanol instead of the sample was used as a blank. GSH was used as a positive control. The DPPH radical scavenging activity of the samples was calculated as follows.
(1)$$\mathrm{DPPH}\;\mathrm{radical}\;\mathrm{scavenging}\;\mathrm{activity}\;\left(\%\right)=\frac{A_{b}-(A_{s}-A_{c)}}{A_{b}}\times 100\%$$
where *As*, *Ac* and *Ab* represent the absorbance of the sample, control and blank groups, respectively.

The IC_50_ value is defined as the concentration of the test sample necessary to achieve 50% clearance. The logistic regression model of SPSS software was used to determine IC_50_ values.

#### 2.4.2. ABTS Radical Scavenging Activity

The ABTS radical scavenging activity of TSP was determined according to the previous method by Yang, X.R. et al., with minor modifications [26]. The GSH and the sample were dispensed into the test solution with concentration gradients of 2, 4, 6, 8 and 10 mg/mL. Then, a stock solution was prepared containing 7 mmol/L ABTS and 2.45 mmol/L potassium persulfate as an ABTS solution and maintained overnight in the dark at room temperature for 15 h to generate ABTS^+^ cations. The ABTS working solution was diluted in distilled water to an absorbance of 0.70 ± 0.02 at 734 nm. Then, 100 µL of sample was added to 400 µL of ABTS working solution and incubated for 10 min at room temperature in the dark, and the absorbance was measured at 734 nm. Distilled water instead of the sample was used as a blank. GSH was adopted as the positive control. The ABTS scavenging activity of the samples was calculated using the following equation.
(2)$$\mathrm{ABTS}\;\mathrm{radical}\;\mathrm{scavenging}\;\mathrm{activity}\;\left(\%\right)=\frac{A_{b}-A_{s}}{A_{b}}\times 100\%$$
where *As* and *Ab* represent the absorbance of the sample and blank group, respectively.

IC_50_ values were calculated by the method described in Section 2.4.1.

### 2.5. Molecular Docking

The three-dimensional structures of DPPH (CID: 2735032) and ABTS (CID: 5360881) were obtained from the PubChem database (https://pubchem.ncbi.nlm.nih.gov/) (accessed on 20 December 2021). The 3D structures of the identified peptides were constructed by Chem Draw 20.0 and Chem 3D software. The interaction between peptide and ligand (DPPH and ABTS) was docked by AutoDock 4.2 software (San Diego, CA, USA), and its conformation was optimized by a genetic algorithm based on the principle of minimizing docking energy. The stereoscopic and planar interactions between the receptor peptide and ligand were analysed by PyMoL (San Carlos, CA, USA) and Ligplot v.2.2.4 software (Cambridge, UK).

### 2.6. In Silico Analysis of the Identified Peptides

The Grand Average of Hydropathy (GRAVY), water solubility, stability, allergenicity and toxicity of the identified peptides were analysed online using ProtParam tool (https://web.expasy.org/protparam/protpar-ref.html) (accessed on 5 August 2022), Innovagen (http://www.innovagen.com/proteomics-tools) (accessed on 24 May 2022), ProtParam tool (http://web.expasy.org/protparam/) (accessed on 24 May 2022), Allergen FP v.1.0 (http://www.ddg-pharmfac.net/AllergenFP) (accessed on 26 May 2022) and ToxinPred (http://crdd.osdd.net/raghava/toxinpred/) (accessed on 26 May 2022), respectively.

### 2.7. Statistical Analysis

All data were analysed by SPSS 17.0 (SPSS Inc., Chicago, IL, USA). One-way ANOVA was used to analyse the differences among the groups; *p* < 0.05 indicates a significant difference and *p* < 0.01 indicates an extremely significant difference.

## 3. Results

### 3.1. Isolation and Purification of Antioxidant Peptides

The TSP with antioxidant capacity was determined with the optimized enzymatic hydrolysis conditions that were obtained from the pre-experiment (pH10, temperature: 51 °C; enzyme dosage: 2.51%, solid–liquid ratio: 1:50 (g/mL)). As shown in Table 1, the IC_50_ values of TSP to the DPPH and ABTS radicals were 6.81 ± 0.06 and 9.69 ± 0.04 mg/mL, respectively. There were various peptides with different molecular weights in TSP, which were separated by ultrafiltration technology under the action of an external driving force (pressure), so fractions with different molecular weights were obtained [27]. Thus, ultrafiltration was used to separate TSP into four fractions (M1–4) based on molecular weight. The fractions were collected after lyophilization, and the antioxidant capacity was determined. As shown in Table 1, fraction M4 exhibited the highest antioxidant capacity, with scavenging rates for the DPPH and ABTS free radicals of 4.61 ± 0.02 mg/mL and 8.43 ± 0.03 mg/mL, respectively, which were significantly higher than that of TSP. The molecular weight of fraction M4 is less than 1000 Da, which belongs to low molecular weight peptides. It was previously shown that the better antioxidant capacity of peptide mixtures is due to the presence of a large proportion of low molecular weight peptides [28]. Therefore, fraction M4 was selected for further experiments.

By utilizing the retention mechanism for the hydrophobic interaction of the solute on the stationary phase, reversed-phase high-performance liquid chromatography (RP-HPLC) has been widely used to further purify active peptides [29]. In this study, RP-HPLC was used to further separate the peptides in fraction M4 and obtain highly active peptides H1–8, as shown in Figure 1a. Each fraction was collected by lyophilization and assayed for antioxidant capacity (Figure 1b,c). The IC_50_ values of the H2 and H8 fractions against the DPPH radical were lower than those of the others, illustrating the higher scavenging activity. Fraction H2 had the highest scavenging activity against DPPH free radicals (IC_50_ = 0.57 ± 0.10 mg/mL), but the yield was low (4.46%). The IC_50_ of H8 against the DPPH free radical was 4.21 ± 0.03 mg/mL, and there was no significant difference in antioxidant capacity between H2 and H8, as indicated by the significance analysis (*p* > 0.05). However, the yield of H8 was the highest (30.89%). It has been shown that purified hydrophilic peptides exhibit certain antioxidant capacity [30]. Fraction H8 was considered to have excellent hydrophobicity, since it was eluted by higher concentrations of organic solvents than those of H2.

The IC_50_ values of H5 and H7 against the ABTS free radical were 3.21 ± 0.01 mg/mL and 2.69 ± 0.58 mg/mL, respectively, with higher scavenging activities than that of the other fractions. Fraction H4, which exhibits the worst ABTS free radical scavenging activity, was still higher than the unpurified fraction M4. The IC_50_ value of fraction H8 was 4.80 ± 0.57 mg/mL, which was not significantly different from that of the H5 component (*p* > 0.05).

The above results showed that fraction H8, which was obtained with the highest yield, showed strong antioxidant capacity as a whole, and the IC_50_ values against DPPH and ABTS radicals were 4.21 ± 0.03 and 4.80 ± 0.57 mg/mL, respectively. Antioxidant peptides from fermented fish (*pekasam*) were previously reported with DPPH and ABTS free radical scavenging activities of 15.31 ± 1.61 and 8.72 ± 1.63 mg/mL, respectively [31]. In contrast, the antioxidant capacity of the purified peptides in this study was improved. Peptides in fraction H8 were eluted by the mobile phase with a higher proportion of organic solvent, indicating a high amount of hydrophobic peptides, which may lead to higher antioxidant capacity [32]. Therefore, the sequence of H8 was identified by LC–MS/MS to further study the relationship between antioxidant capacity and peptide structure.

### 3.2. Identification of Antioxidant Peptides by LC–MS/MS

In LC–MS/MS, liquid chromatography separates the analytes and interfering substances, and mass spectrometer is used for detection. MS, which has high sensitivity, selectivity and accuracy, is the main technical means used to identify the purified active peptide sequence [33]. Twenty novel peptides were identified in fraction H8, and the amino acid sequences, length and molecular weight are summarized in Table 2. The twenty novel peptides were synthesized by GL Biochem Ltd. (Shanghai, China) by a solid-phase peptide method to evaluate their antioxidant capacity and scavenging activities against DPPH and ABTS. As shown in Figure 2, six synthetic peptides, namely, P1 (GGYDEY), P2 (GGYDEYR), P3 (KAPDPGPGPM), P4 (GNAGPTGPAGPL), P5 (GAPGARGPNGY) and P6 (SGPPVPGPIGPM), were found to be active against DPPH and ABTS. The above six peptides were characterized by HPLC and MS (Appendix A).

P4 exhibited higher DPPH radical scavenging activity in all peptides, with IC_50_ values of 3.69 ± 0.30 mg/mL and 2.56 ± 0.15 mg/mL, respectively, which were slightly higher than that of the original H8 fraction (IC_50_ = 4.21 ± 0.03 mg/mL). P1 showed the highest ABTS free radical scavenging activity, which was lower than that of the original fraction H8, but there was no significant difference (*p* > 0.05). The molecular weight range of 702~1104 Da of the selected peptides is consistent with the better biological activity of low molecular weight peptides that was observed in previous studies [34].

The antioxidant capacity of peptides is influenced by their structural characteristics, especially their molecular mass, amino acid composition and sequence [35]. The number of amino acids was thought to affect the free radical scavenging activity of active peptides. Active peptides derived from animals are mostly composed of 4–12 amino acids, contain molecular weights ranging from 400.4 to 1153.4 Da and show excellent antioxidant capacity [36,37,38,39]. The molecular weights of P1 (6 residues), P3 (10 residues) and P4 (12 residues) are 702.2, 966.1 and 1007.5 Da, respectively (Table 2), which results in satisfactory antioxidant capacity. The amino acid composition of a peptide is also critical for its biological activity. The presence of hydrophobic amino acids is believed to greatly contribute to the enhancement of the antioxidant capacity of the peptides [40,41]. Peptides containing hydrophobic amino acids, such as Pro and Ala, were identified in screened active peptides P1, P3 and P4. The GRVAY value is calculated as the sum of the hydrophilicity values of all amino acids divided by the number of residues in the sequence and can be used to estimate the hydrophobicity or hydrophilicity of a peptide sequence [42]. Cavaliere, C. et al. performed a predictive analysis of the identified antioxidant peptides and obtained a GRAVY index ranging between −2 and +2 [43]. According to Table 2, P1, P3, P4, P5 and P6 all showed GRAVY values within the range, while the P2 value was only slightly above +2. At the same time, P6 showed a positive value, and other peptides showed a negative value but close to 0, indicating that these sequences all have a certain degree of hydrophobicity, or can be characterized as amphiphilic peptides. In addition, the position of amino acids in the peptide sequence also has a great influence on antioxidant capacity [44]. The presence of Gly at the N-terminus is considered to be the general structure of fish skin collagen peptides [45], and there is evidence that aromatic amino acids (Tyr) at the C-terminus contribute to the enhancement of the antioxidant capacity of the peptide [46]. Both P1 and P5 contain Tyr as the C-terminal residue, which may be closely related to their potent antioxidant capacity. Leu-Ser-Gly-Tyr-Gly-Pro (LSGYGP) and Gly-Leu-Phe-Gly-Pro-Arg (GLFGPR), which are also obtained from fish skin, have been identified to have exciting antioxidant capacity [23,36]. The identification of the GP sequences in the above peptides may have a positive effect on the antioxidant capacity of TSP. The above analysis shows that Pro, Ala and Tyr are the characteristic amino acids in the free radical scavenging reaction of TSP.

### 3.3. Antioxidant Mechanism of Peptides by Molecular Docking

#### 3.3.1. Docking and Interaction Model between P1–P6 and DPPH

Molecular docking technology has been widely used to study the interaction mechanism between antioxidant peptides and free radicals [16]. The optimal binding configurations of P1–P6 to DPPH were based on the minimum binding energy after docking. The minimum binding energies of P1–P6 were −1.96, −2.53, −1.38, −2.25, −0.94 and −1.89 kcal/mol, respectively, and P2 had the lowest binding energy compared to that of the other peptides.

Combined with Figure 3 and Table 3, hydrophobic interactions existed between all amino acid residues of P1 and DPPH, with the strongest interaction at the Asp-4 residue. A hydrogen bond was formed between the Tyr-6 residue of P2 and DPPH, and this bond was accompanied by hydrophobic interactions. At the same time, Gly-2, Tyr-3, Asp-4 and Glu-5 residues also formed hydrophobic interactions with ligands. Residues Pro-5, Pro-9, Lys-1 and Gly-6 in P3 formed hydrophobic interactions with DPPH, and the Pro-5 residue had the strongest hydrophobic interaction. Nine amino acid residues in P4 bound to the ligand species through hydrophobic interactions (Figure 3d). In addition, a hydrogen bond was formed between the Thr-6 residue of P4 and DPPH. A hydrogen bond with a hydrogen bond length of 2.80 Å was present between the ligand and the Arg-6 residue in P5. In addition, the residues Arg-6, Ala-5, Gly-7 and Pro-8 jointly provided hydrophobic interactions between the receptor and ligand. Notably, the Gly-7 residue in P6 also formed hydrophobic interactions with the DPPH and Val-5, Pro-6, Pro-8 and Ile-9 residues, while providing a hydrogen bond with a bond length of 2.72 Å.

Currently, hydrogen atom transfer (HAT) is considered to be the main mechanism for scavenging DPPH. As a hydrogen donor, the peptide pairs with the single electron of DPPH that exists in the form of free radicals in solution to complete the free radical scavenging [47]. The six identified peptides are rich in amino acid residues such as Pro, Tyr and Ala, which are reported to be effective hydrogen donors and play an important role in the antioxidant capacity of peptides [48,49]. Based on the simulation results, the identified peptides and DPPH molecules had strong hydrophobic interactions and hydrogen bonds, so the peptides had strong DPPH free radical scavenging activity. It is widely believed that the presence of hydrogen bonds and hydrophobic interactions is closely related to the free radical scavenging ability of active peptides [50,51]. P4 showed the best scavenging activity against DPPH free radicals among all peptides (IC_50_ = 2.56 ± 0.15 mg/mL). This benefited from the presence of hydrogen bonds between P4 and DPPH, and P4 also contains the most amino acid residues for hydrophobic interactions with ligands. The carboxyl group in the Tyr-6 residue in P1 bound with DPPH through hydrophobic interactions, and the free radical scavenging activity improved as a result. It has been reported that the salmon-derived antioxidant peptide PMRGGGGYHY, which is obtained by molecular docking, contains a Tyr residue as the main antioxidant active site. Hydrogen bonding significantly increased the antioxidant capacity [46]. The strong hydrogen bonds and hydrophobic interactions are extremely satisfactory between the antioxidant peptides TSSSLNMAVRGGLTR and STTVGLGISMRSASVR from millet digestate and the target molecules DPPH and ABTS [52]. The presence of aromatic amino acids at the C-terminus could previously enhance the antioxidant capacity of peptides; this occurs because the aromatic amino acids cause the frontier orbital (HOMO-LUMO) energy gap (the difference between the energies of the lowest point of the conduction band and the highest point of the valence band) to be smaller for the intermediate residues, which has been demonstrated by quantitative structure–activity relationship (QSAR) analysis [53]. Tyr was located at the C-terminus of P1 and P5 in this study, which may have a positive effect on antioxidant capacity. P4 exhibited the best DPPH free radical scavenging activity. The Thr-6 residue provided the active site for hydrogen bonding for the ligand, and Ala, Gly and Pro provided the main site for hydrophobic interactions. In summary, the existence of Pro, Tyr, Ala and Thr residues has a greater contribution to scavenging free radicals, which is considered from the analysis of the docking results of DPPH and peptide molecules.

#### 3.3.2. Docking and Interaction Model between P1–P6 and ABTS

The optimal binding configuration of P1–P6 and ABTS was obtained based on the minimum binding energy after docking, −4.28, −3.44, −2.27, −3.77, −2.26 and −3.13 kcal/mol, respectively. P1 had the lowest binding energy compared to that of other peptides. Combined with Figure 4 and Table 4, two hydrogen bonds were formed between the Gly-1 and Glu-5 residues of P1 and ABTS, respectively. Tyr (Tyr-3, Tyr-6) and Asp-4 residues of P1 formed hydrophobic interactions with ABTS. The Glu-5 residue of P2 formed a hydrogen bond of 2.57 Å bond length with ABTS, and there were hydrophobic interactions between Tyr (Tyr-3, Tyr-6), Gly-2 and Asp-4 residues and ABTS. The Lys-1, Pro (Pro-5, Pro-7, Pro-9) and Gly (Gly-6, Gly-8) residues of P3 had hydrophobic interactions with ABTS, and the interaction with Pro-7 residues was the strongest. Two hydrogen bonds were formed between amino residues (Thr-6) of P4 and ABTS. The ligand ABTS also exhibited hydrophobic interactions with Ala (Ala-3, Ala-9), Gly (Gly-4, Gly-10), Pro (Pro-8, Pro-11) and Leu-12 residues of P4. In addition, the Tyr-11 residue of P5 formed a hydrogen bond with ABTS with a bond length of 2.68 Å. The Pro-3, Ala-5, Asn-9 and Gly-10 residues of P5 also have hydrophobic interactions with the ligands. The Pro (Pro-4, Pro-6, Pro-8), Val-5, Gly-7 and Ile-9 residues of P6 formed hydrophobic interactions with ABTS. The Pro-6 residue exhibited the strongest hydrophobic interaction with ABTS.

It has been reported that electron transfer (ET) is the main mechanism of ABTS^+^ scavenging, which changes the oxidation state of substances through electron movement between peptides and free radicals [54]. Gly residues located on the side chain can be neutralized by free radicals; this is achieved through donating protons due to the presence of individual hydrogen atoms [55]. The P1 obtained in this study had excellent antioxidant capacity, and this was due to the Gly residue at the N-terminus, which formed a hydrogen bond with the ligand ABTS. At the same time, another hydrogen bond between Glu and ABTS was provided, so P1 exhibited the highest scavenging rate of ABTS free radicals (IC_50_ = 9.14 ± 0.08 mg/mL). The DTETGVPT derived from False Abalone also exerted an excellent antioxidant effect in vivo due to the presence of hydrogen bonds [56]. It was shown that the scavenging of ABTS free radicals by Thr and Tyr was mainly due to the hydroxyl groups on phenol in some studies. The semiquinone radical was formed by donating hydrogen to the ABTS radical through the hydroxyl group, and then the peptide molecule that lost one hydrogen atom reacted with another ABTS radical, leading to the formation of polyphenol-derived adducts. Free radicals are scavenged due to the instability and fragmentation of these polyphenol-derived adducts [15]. P4 and P5 in this study presented excellent antioxidant capacity because they formed strong hydrogen bonds with ligands via Thr and Tyr residues, respectively. In addition, as shown in Figure 4c,f, the Pro residue provided the main hydrophobic interaction with free radicals, which provided the main active site for both the P3 and P6 peptides. Studies have shown that stronger hydrophobic interactions have a positive effect on the scavenging of ABTS^+^ free radicals. In the presence of Pro, the EC_50_ value of VGPWQK was 1.00 ± 0.00 mg/mL, which was comparable to that of the commercial antioxidant GSH (0.69 mg/mL) [57]. P1 was the best substance for ABTS free radical scavenging activity, Gly-1 and Glu-5 provided two hydrogen-bonding active sites for ligands and Tyr and Asp were the main amino acids that provided hydrophobic interactions. In general, the presence of Gly, Thr, Glu, Tyr and Pro residues contributed more to the scavenging of ABTS free radicals by the peptide molecules demonstrated by the analysis of the docking results.

### 3.4. In Silico Analysis of Identified Peptides from Tilapia Skin

The water solubility, stability, allergenicity and toxicity of the six synthetic peptides were predicted and analysed by computer analysis (Table 5). Good water solubility and stability were predicted in P2, P3 and P5, which was beneficial to maintaining their biological activity during processing and digestion. In terms of food safety, all peptides except for P1 and P2 are considered probable nonallergens. In addition, all peptides were determined to be nontoxic. It was comprehensively shown that the good processing performance and safety were reflected in the two antioxidant peptides P3 and P5, which exhibited good free radical scavenging activity. On the other hand, P1 (ABTS^+^) and P4 (DPPH), which exhibit excellent scavenging activity, were considered to be deficient in allergenicity and water solubility, respectively. Therefore, the peptides screened in this study are suitable for application in the food industry as antioxidants.

## 4. Conclusions

In this study, six new peptides (P1–P6) were identified from the enzymatic hydrolysate of tilapia skin, and the structure–activity relationship of TSP was investigated by molecular docking. High antioxidant capacity was found due to their low molecular weight and the presence of hydrophobic (Pro, Ala) and aromatic amino acids (Tyr) in the sequence. These peptides also acted as the functional characteristic amino acids in TSP free radical scavenging (DPPH, ABTS) reactions. Molecular docking revealed that the peptides were able to bind to free radicals by hydrogen bonds and hydrophobic interactions, thereby inhibiting free radicals. These results could also clarify the key active sites of P1–P6 and their structure–activity relationship. Among them, P4 and P1 exhibited the strongest DPPH and ABTS free radical scavenging activities, respectively, which can interact with free radicals through hydrogen bonds and hydrophobic interactions. It was also revealed that the hydrogen-bonding active site of P4 was located at the Thr-6 residue, and Ala, Gly and Pro were amino acids that provided hydrophobic interactions. In addition, the Gly-1 and Glu-5 residues of P1 provided two hydrogen-bonding active sites, and Tyr and Asp were the main amino acids that provided hydrophobic interactions. Overall, the two identified peptides with high antioxidant capacity can be used as natural antioxidants. Furthermore, the characteristic amino acids and interaction mechanism during the binding process of peptides and free radicals were determined, which provided new ideas for the mechanistic research and application design of tilapia skin antioxidant peptides, as well as their development and utilization in the food industry.

## Data Availability

Data are contained within the article.

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
