# Peer review of "Identification of Antioxidant Peptides Derived from Tilapia (Oreochromis niloticus) Skin and Their Mechanism of Action by Molecular Docking"

_foods, 2022, doi:10.3390/foods11172576_

Round 1

Reviewer 1 Report

In the present study the authors isolated and identified six antioxidant peptides from the enzymatic hydrolysate of tilapia skin based on the free radical scavenging ability of the fractions to DPPH and ABTS assay. Molecular docking were then conducted to study the interaction mechanism between antioxidant peptides and free radicals.

Major comments:

11) Please provide the justification of the use of DPPH and ABTS assay for the evaluation of antioxidant reactivity here. One major limitation of the spectrometric DPPH assay is the overlapped spectra of fractions that absorb in the same wavelength as DPPH, that might interfere with the results and interpretations. Please comment on this.

22) The differences in the antioxidant activity of IC50 of the fractions (M1-M4) measured by DPPH and ABTS (as shown in Table 1) are very different suggesting a major flaw in these assays. M2 and M3 fractions give ~1.4-fold differences antioxidant activity but this fold-change was not observed with ABTS assay. There are hardly any differences between the fractions in the ABTS assay and the trend of the synthetic peptides (P1-P6) in both assays DPPH and ABTS (Figure 2) were very different, please explain this. 

33) DPPH assay’s sensitivity may be affected by the presence and concentration of hydrogen and metal ion and freshness of DPPH. Please explain how the author minimizes the impact of other interferences.

44) Unless I missed this in the manuscript, the characterization of purified, isolated/synthesized peptides (P1-P6) by mass spectrometry and NMR spectroscopy should be included in the manuscript.

Minor comments:

Please comment/suggest briefly on how the future validation on the key active sites for inhibiting free radicals can be conducted.

Reviewer 2 Report

This paper presents an interesting study on the antioxidant activities of the hydrolysates from Tilapia skin, which are generated as by-products from fish processing industry. The authors produced the hydrolysates according to an optimized enzymatic treatment, recovering the fraction below 1 kDa by dead-end ultrafiltration. The sequential purification by reversed-phased chromatography, followed by Mass Spectrometry, allowed the identification of 20 peptide species, among wich 6 presented strong DPPH and ABTS in vitro activities. The antioxidant activity, as well as other features such as solubility or toxicity, were analysed by computer tools (molecular docking, toxinPred, etc).

This work provides a complete study joining in vitro experimentation with computer simulation tools, useful to identify the molecular species responsible for the antioxidant properties of Tilapia skin hydrolysates.  I append a pdf file containing all my comments on the manuscript.

Reviewer 3 Report

The work described in this manuscript is very interesting and relevant, given the popularity of Oreochromis niloticus in the global market and the high availability of wastes of this fish. These are my comments:

-The authors indicate in the abstract that the identified/characterized peptides might be used in the food industry. They are right. However, this potential use of peptides is not indicated/developed in the introduction section. They indicated in the introduction section the negative effects of free radicals on human health, however, the intestinal tract cannot absorb large peptides as those studied by the authors and therefore these peptides might not exert beneficial properties in intracellular environments. Human intestine can only absorb free amino acids and di- and tri-peptides.  

- Scheme 1 is a little bit confusing for me. It should be improved or deleted.

-The authors indicated the composition of buffer B used for the LC-MS/MS analysis, however, they did not indicate the composition of solvent A.  

-I missed the quantities of GSH and peptides used in the antioxidant capacity assays.

-The authors should indicate the methodology used for yield evaluation. I missed the yield for TSP in Table 1.

-The authors identified 20 new peptides but only reported the antioxidant capacity of 6 of them.  Why? I missed the antioxidant capacity of the other peptides. All peptides have antioxidant capacity.

-The authors talk about the antioxidant activity of peptides. However, the methods used by the authors to study this property of peptides are classified as methods to evaluate the antioxidant capacity. Antioxidant capacity and antioxidant activity are two different concepts. Did you evaluate the antioxidant capacity or antioxidant activity? I believe that you studied the antioxidant capacity.

-Please, review the following sentence “P3 and P4 exhibited higher DPPH radical scavenging activity in all peptides,” It does not make sense for me.
